# Synthesis and Biological Activity of Benzamides Substituted with Pyridine-Linked 1,2,4-Oxadiazole

**DOI:** 10.3390/molecules25153500

**Published:** 2020-07-31

**Authors:** Sen Yang, Xiao-Yu Tian, Tian-Yang Ma, Li Dai, Chao-Li Ren, Jun-Chang Mei, Xing-Hai Liu, Cheng-Xia Tan

**Affiliations:** College of Chemical Engineering, Zhejiang University of Technology, Hangzhou 310014, China; y17764528124@163.com (S.Y.); t895745887@163.com (X.-Y.T.); m15755187338@163.com (T.-Y.M.); 15157357729@163.com (L.D.); 18708589843@163.com (C.-L.R.); meijunchang_ucb@163.com (J.-C.M.); xhliu@zjut.edu.cn (X.-H.L.)

**Keywords:** amide compound, fungicidal activity, insecticidal activity, 1,2,4-oxadiazole, synthesis

## Abstract

To find pesticidal lead compounds with high activity, a series of novel benzamides substituted with pyridine-linked 1,2,4-oxadiazole were designed by bioisosterism, and synthesized easily via esterification, cyanation, cyclization and aminolysis reactions. The structures of the target compounds were confirmed by ^1^H-NMR, ^13^C-NMR and HRMS. The preliminary bioassay showed that most compounds had good larvicidal activities against mosquito larvae at 10 mg/L, especially compound **7a**, with a larvicidal activity as high as 100%, and even at 1 mg/L was still 40%; at 50 mg/L, all the target compounds showed good fungicidal activities against the eight tested fungi. Moreover, compound **7h** exhibited better inhibitory activity (90.5%) than fluxapyroxad (63.6%) against *Botrytis cinereal*. Therefore, this type of compound can be further studied.

## 1. Introduction

Recently, the demand for pesticides has been increasing with the improvement of people’s living standards. Many new kinds of pesticides with high efficiency, low toxicity and low residue have been developed. Heterocyclic derivatives received important attention due to their various biological activities [1,2,3,4,5,6]. Among these compounds, 1,2,4-oxadiazole heterocyclic derivatives, which contain nitrogen and oxygen atoms, displayed diverse activities, such as insecticidal activity [7,8,9], antifungal activity [10,11], herbicidal activity [12], anti-inflammatory [13], hypotensive [14] and other physiological activity. In addition, amide groups could easily generate hydrogen bonds with the activated part of the target enzymes and control the target organisms. Moreover, introducing the amide structure was beneficial for the biodegradation of pesticides. So, amide derivatives were wildly studied [15,16,17,18,19,20], such as the commercial herbicide flufenacet (Figure 1), insecticide flubendiamide (Figure 1) and fungicide flutolanil (Figure 1).

Amide compounds could be roughly classified as carboxyamides, mandelic acid amides and phenylamides according to their chemical structures. Through analyzing the reported amide compounds, it was found that when the carboxylic acid parts of the compounds include the structural units of pyridine-linked heterocycle, amide compounds usually have better insecticidal or fungicidal activities. Most of the compounds that have been reported include the structural units of pyridine-linked pyrazole, such as the commercial insecticide chlorantraniliprole (Figure 1) and the compound **A** (Figure 1, at 50 mg/L, its fungicidal activity against *Botrytis cinerea* was 80%) synthesized by Xu et al. [21]. However, there are few reports about pyridine-linked 1,2,4-oxadiazole. Moreover, Syngenta has reported a lot of 1,2,4-oxadiazole substituted benzamides such as compounds **B** (Figure 1) and **C** (Figure **1**) that had good fungicidal activities [22,23]. In view of these facts mentioned above, to find pesticides with high biological activity, using **A** and **B** as lead compounds and replacing the structure of trifluoromethyl with the pyridine ring, a series of novel benzamides substituted with pyridine-linked 1,2,4-oxadiazole have been designed and synthesized according to bioisosterism (Figure 2). Target compounds were confirmed by ^1^H-NMR, ^13^C-NMR and HRMS. Their insecticidal and fungicidal activities were studied and the result showed that the target compounds had good insecticidal and fungicidal activities. The synthetic route of the target compounds is shown in Scheme 1.

## 2. Results and Discussion

### 2.1. Synthesis of Target Compounds

The starting material 2-chloro-5-iodobenzoic acid **1** experienced esterification and cyanation reaction to give methyl 2-chloro-5-cyanobenzoate **3**. Methyl 2-chloro-5-cyanobenzoate **3** then reacted with NH_2_OH**•**HCl under the alkaline condition to give amine oxime. This reaction could not be carried out for a long time, because amide byproducts would have formed easily. After this step, 2-chloro-5-(5-(3,6-dichloropyridin-2-yl)-1,2,4-oxadiazol-3-yl) benzoate **5** was synthesized by cyclization reaction from compounds **4** with 3,6-dichloropicolinoyl chloride that had been synthesized from 3,6-dichloropicolinic acid. At last, the intermediate **5** went through hydrolysis reaction, and then reacted with substituted aniline to give the target compound **7**.

In step 2, the iodine atom on the benzene ring being replaced by a cyano group is a typical nucleophilic substitution reaction. NaCN and KCN are two common nucleophilic reagents. In fact, these two reagents are highly toxic, so we chose CuCN, which has relatively low toxicity, as the cyanidation reagent to reduce the risk in the experiment and the harm to the environment. In addition, the process of our experiment was different from that reported in former papers [24,25]. Using small natural organic molecule l-proline as the catalyst, DMF as the solvent and under the condition where temperature gradually increased, the amount of by-product decreased and the yield of product **3** was the highest [26]. The influence of different experimental conditions on the yield of product **3** is shown in Table 1.

In step 4, the formation of 1,2,4-oxadiazole was achieved in a one-pot reaction. 3,6-dichloropicolinoyl chloride that was freshly prepared was dropped into the solution of methyl-2-chloro-5-(*N*′-hydroxycarbamimidoyl) benzoate **4** and triethylamine in toluene at 0 °C, to give the intermediate of methyl-2-chloro-5-(*N*-(3,6-dichloropicolinoyl)-*N*′-hydroxycarbamimidoyl)benzoate. Next, the intermediate was cyclized to produce 1,2,4-oxadiazole at reflux (Scheme 2). In this way, the self-cyclization of compound **4** was avoided because of the higher reactivity of acid chloride. 

### 2.2. Spectrum Analysis of Target Compounds

The target compound **7a** was taken as an example to conduct spectrum analysis. In the ^1^H-NMR spectra of compound **7a**, the singlet at δ 10.09 ppm was the NH peak. The signals of benzene and pyridine rings were assigned at 8.37–7.04 ppm. In the ^13^C-NMR spectra of compound **7a**, the C=O signal could be found at 171.75 ppm. The appearance of signals at 167.27 and 163.91 ppm was assigned to the carbons of the 1,2,4-oxadiazole ring. In the HRMS spectrogram, the calculated value of the ion peak of this compound was [M + Na]^+^ 466.9840, and the measured value was [M + Na]^+^ 466.9840. The absolute error was within 0.003.

### 2.3. Biological Activities of Target Compounds

The results of the insecticidal activity test of the target compounds are shown in Table 2 and Table 3. In Table 2, the death rates of compound **7** are all below 50% against *mythimna sepatara*, *helicoverpa armigera* and *pyrausta nubilalis* at 500 mg/L. Therefore, the insecticidal activities of this series of compounds were not good enough against the three targets. Nevertheless, we can see that compound **7** exhibited good larvicidal activity against mosquito larvae from Table 3. The larvicidal activities of compounds **7a** and **7f** were 100% at 10 mg/L. Furthermore, it was found that the larvicidal activity of compound **7a** was 100% at 2 mg/L. Even at 1 mg/L, the larvicidal activity was still 40%. It exhibited better larvicidal potency than etoxazole against mosquito larvae. It can be seen from the compound **7h**, **7i** and **7j** that the position of the substituent has little effect on the larvicidal activity of the target compound. Furthermore, as for compounds **7a** to **7n**, from the general trend in the larvicidal activity, it can be concluded that the less steric substitution attached to aniline may reduce the obstacles of the target compound binding to the target receptor and help to bring about an increase in activity.

The results of the fungicidal activities of target compounds are shown in Table 4. Overall, among the 14 new compounds, only compound **7h** showed good fungidical activities (90.5%) against *Botrytis cinereal*, which was better than fluxapyroxad (63.6%). At the same time, it had moderate inhibitory activities against *Alternaria solani* (50.0%), *Sclerotinia sclerotiorum* (80.8%), and *Thanatephorus cucumeris* (84.8%). In addition, compounds **7d** and **7e** also showed good fungicidal activities against *Botrytis cinereal* (66.7%, 63.1%), which was comparable to fluxapyroxad. Most compounds had moderate inhibitory activities (40–81%) against *Sclerotinia sclerotiorum*, such as compounds **7b**, **7e**, **7h**, **7i**, **7j**, **7l** and **7n**. For *FusaHum graminearum*, *Phytophthora capsica* and *Fusarium oxysporum*, fluxapyroxad had almost no activities, while compounds **7e**, **7h**, **7m** had moderately strong inhibitory activities against *FusaHum graminearum* (40–45%), and the other two fungi had weak activity. From Table 5, we can see that compound **7h** had good inhibitory activity against *Sclerotinia sclerotiorum*, *Botrytis cinereal* and *Thanatephorus cucumeris*, with EC_50_ of 11.61, 17.39 and 17.29 μg/mL, respectively. The SAR of the target compounds about fungicidal activities (Table 4) is that when the substituent of the benzene ring is 2-F, the inhibitory activities against the tested fungi are superior to other compounds.

## 3. Experimental Section

### 3.1. General Information

Melting points were determined using an X-4 digital microscopic melting point detector (Taike, Beijing, China) and the thermometer was uncorrected. ^1^H-NMR and ^13^C-NMR spectra were measured on NMR spectrometer (Bruker 500 MHz, Fallanden, Switzerland); High-resolution electrospray mass spectra (HR–ESI–MS) were determined using an UPLC H-CLASS/QTOF G2-XS mass spectrometer (Waters, Milford, MA, USA). All the reagents and solvents were in analytical purity. The characterisation data for all synthesised compounds are provided in the Appendix A.

### 3.2. Synthesis

#### 3.2.1. Methyl 2-chloro-5-iodobenzoate **2**

2-chloro-5-iodobenzoic acid **1** (2.8 g, 0.01 mol), methanol (50 mL) and H_2_SO4 (0.5 mL) were added to a three-necked flask and reacted at reflux for about 8 h. After the mixture was cooled to room temperature, methanol was removed under reduced pressure. Then, EtOAc (50 mL) was added and the pH adjusted to 7–8 by using NaHCO_3_. The organic layer was dried with Na_2_SO_4_. The solvent was removed to give 2.71 g creamy white solid. Yield: 92%, m.p. 40~42 °C; ^1^H-NMR (500 MHz, Chloroform-*d*) δ 8.14 (d, *J* = 2.0 Hz, 1H), 7.72 (dd, *J* = 8.5, 2.5 Hz, 1H), 7.19 (d, *J* = 8.0 Hz, 1H), 3.94 (s, 3H).

#### 3.2.2. Methyl 2-chloro-5-cyanobenzoate **3**

Methyl 2-chloro-5-iodobenzoate **2** (0.40 g, 1.3 mmol), CuCN (0.18 g, 2.0 mmol), l-proline (0.15 g, 1.3 mmol) and DMF (15 mL) were added to a three-necked flask. After being dissolved, the mixture was heated to 70 °C and reacted for 2 h. Then, the temperature was heated to 100 °C. The reaction was complete after 9 h. After the mixture was cooled to room temperature, it was filtered with diatomite. The filtrate was extracted by water (100 mL) and EtOAc (100 mL). The organic layer was washed with water (50 mL × 3) and then dried with MgSO_4_. EtOAc was removed under reduced pressure and 0.20 g yellow solid was obtained. Yield: 79%, m.p. 102–104 °C ([27], 100–103 °C).

#### 3.2.3. Methyl 2-chloro-5-(*N*′-hydroxycarbamimidoyl)benzoate **4**

Methyl 2-chloro-5-cyanobenzoate **3** (1.4 g, 7.2 mmol) was added to a three-necked flask and dissolved by ethanol (45 mL). Stirring was started at room temperature, and then hydroxylamine hydrochloride (0.75 g) and triethylamine (1.1 g) were gradually added. The mixture was stirred for 3 h. Then the solvent was removed under reduced pressure and the remnant was dissolved in EtOAc (50 mL) and saturated saline (50 mL). The organic layer was dried with Na_2_SO_4_ and evaporated, to give 1.5 g light yellow solid. Yield: 90%, m.p. 108–111 °C; ^1^H-NMR (500 MHz, DMSO-*d*_6_) δ 9.87 (s, 1H), 8.13 (d, *J* = 2.0 Hz, 1H), 7.85 (dd, *J* = 8.5, 2.0 Hz, 1H), 7.59 (d, *J* = 8.5 Hz, 1H), 5.98 (s, 2H), 3.87 (s, 3H).

#### 3.2.4. Methyl 2-chloro-5-(5-(3,6-dichloropyridin-2-yl)-1,2,4-oxadiazol-3-yl)benzoate **5**

3,6-dichloropicolinic acid (1.1 g, 5 mmol) and SOCl_2_ (20 mL) were added to a round bottom flask. The mixture was stirred and refluxed for 2 h. Then, SOCl_2_ was removed under reduced pressure to give 3,6-dichloropicolinoyl chloride.

Intermediate **4** (1.1 g, 5 mmol), triethylamine (1.2 g, 5 mmol) and dry toluene (100 mL) were added to a three-necked flask. The mixture was stirred at 0 °C for 2 h. After that, the prepared 3,6-dichloropicolinoyl chloride (dissolved by 30 mL dry toluene) was dropped into the flask. The mixture continued to be stirred for 1 h at 0 °C. Then, the temperature was increased to reflux for 2 h. When the mixture was cooled to room temperature, it was washed by saturated sodium chloride solution (150 mL × 3). The organic layer was dried by Na_2_SO_4_ and removed under reduced pressure to give 1.5 g yellow solid. Yield: 78%, m.p. 148–152 °C; ^1^H-NMR (500 MHz, Chloroform-*d*) δ 8.65 (s, 1H), 8.32–8.16 (m, 1H), 7.92 (d, *J* = 8.5 Hz, 1H), 7.61 (d, *J* = 8.5 Hz, 1H), 7.54 (d, *J* = 8.5 Hz, 1H), 3.98 (s, 3H).

#### 3.2.5. 2-chloro-5-(5-(3,6-dichloropyridin-2-yl)-1,2,4-oxadiazol-3-yl)benzoic acid **6**

Intermediate **5** (0.8 g, 2.0 mmol) and THF (40 mL) were added to a three-necked flask. After being dissolved, 30% NaOH (5 mL) was also added to the flask and refluxed for 2 h. After the mixture was cooled to room temperature, the solvent was removed. Then, the pH was adjusted to 2–3 with HCl and 0.7g white solid precipitate was obtained. Yield: 94%, m.p. 203–204 °C; ^1^H-NMR (500 MHz, DMSO-*d*_6_) δ 13.64 (s, 1H), 8.45 (d, *J* = 2.1 Hz, 1H), 8.34 (s, 1H), 8.26–8.17 (m, 1H), 7.91 (d, *J* = 8.6 Hz, 1H), 7.80 (d, *J* = 8.3 Hz, 1H).

#### 3.2.6. Preparation of Target Compound **7**

Intermediate **6** (4.1 g, 11 mmol), triethylamine (0.2 g), DCM (30 mL) and EDCI (0.3 g) were added to a three-necked flask. Substituted aniline (12 mmol) was stirred and dropped into the flask at 0 °C. TLC was used to track reaction progress. At last, target compound **7** was obtained by using the method of column chromatography separation.

*2-chloro-5-(5-(3,6-dichloropyridin-2-yl)-1,2,4-oxadiazol-3-yl)-N-phenylbenzamide***7a**. Yellow solid, yield 64%, m.p. 234–238 °C, ^1^H-NMR (500 MHz, DMSO-*d*_6_) δ 10.70 (s, 1H), 8.36 (d, *J* = 8.5 Hz, 1H), 8.25–8.18 (m, 2H), 7.92 (d, *J* = 9.0 Hz, 1H), 7.85 (d, *J* = 8.0 Hz, 1H), 7.73 (d, *J* = 8.0 Hz, 2H), 7.38 (t, *J* = 7.5 Hz, 2H), 7.19–7.09 (m, 1H); ^13^C-NMR (500 MHz, DMSO-*d*_6_) δ 171.75, 167.27, 163.91, 148.60, 143.34, 139.71, 138.69, 137.84, 133.65, 131.28, 131.10, 129.52, 129.27, 128.88, 127.32, 124.77, 124.13, 119.78; HRMS calcd. for C_20_H_11_Cl_3_N_4_O_2_ [M + Na]^+^ 466.9840, found 466.9840.

*2-chloro-5-(5-(3,6-dichloropyridin-2-yl)-1,2,4-oxadiazol-3-yl)-N-(o-tolyl)benzamide***7b**. Yellow solid, yield 77%, m.p. 220–223 °C, ^1^H-NMR (500 MHz, DMSO-*d*_6_) δ 10.60 (s, 1H), 8.34 (d, *J* = 9.0 Hz, 1H), 8.21 (d, *J* = 7.0 Hz, 2H), 7.91 (d, *J* = 9.0 Hz, 1H), 7.83 (d, *J* = 9.0 Hz, 1H), 7.61 (d, *J* = 8.5Hz, 2H), 7.17 (d, *J* = 8.5 Hz, 2H), 2.29 (s, 3H); ^13^C-NMR (500 MHz, DMSO-*d*_6_) δ 171.72, 167.26, 163.68, 148.59, 143.31, 139.68, 137.90, 136.19, 133.65, 133.12, 131.25, 131.07, 129.43, 129.24, 129.21, 127.31, 124.73, 119.76, 20.52; HRMS calcd. for C_21_H_13_Cl_3_N_4_O_2_ [M + Na]^+^ 480.9996, found 480.9995.

*2-chloro-5-(5-(3,6-dichloropyridin-2-yl)-1,2,4-oxadiazol-3-yl)-N-(m-tolyl)benzamide***7c**. Yellow solid, yield 74%, m.p. 260–261 °C, ^1^H-NMR (500 MHz, DMSO-*d*_6_) δ 10.69 (s, 1H), 8.35 (d, *J* = 8.5 Hz, 1H), 8.25–8.17 (m, 2H), 7.92 (d, *J* = 9.0 Hz, 1H), 7.83 (d, *J* = 9.0 Hz, 1H), 7.60 (s, 1H), 7.51 (d, *J* = 8.5 Hz, 1H), 7.25 (t, *J* = 8.0 Hz, 1H), 6.96 (d, *J* = 7.5 Hz, 1H), 2.32 (s, 3H); ^13^C-NMR (500 MHz, DMSO-*d*_6_) δ 171.73, 167.27, 163.88, 148.60, 143.34, 139.70, 138.63, 138.10, 137.90, 133.65, 131.27, 131.08, 129.47, 129.27, 128.69, 127.30, 124.80, 124.74, 120.29, 117.01, 21.22; HRMS calcd. for C_21_H_13_Cl_3_N_4_O_2_ [M + Na]^+^ 480.9996, found 480.9998.

*2-chloro-5-(5-(3,6-dichloropyridin-2-yl)-1,2,4-oxadiazol-3-yl)-N-(p-tolyl)benzamide***7d**. White solid, yield 68%, m.p. 230–232 °C, ^1^H-NMR (500 MHz, DMSO-*d*_6_)δ 10.19 (s, 1H), 8.36 (d, *J* =10 Hz, 1H), 8.27 (d, *J* = 1.9 Hz, 1H), 8.24–8.18 (m, 1H), 7.93 (d, *J* = 8.5 Hz, 1H), 7.84 (d, *J* = 8.4 Hz, 1H), 7.50 (d, *J* = 7.7 Hz, 1H), 7.29 (d, *J* = 7.3 Hz, 1H), 7.25 (t, *J* = 7.3 Hz, 2H), 7.20–7.16 (m, 1H), 2.32 (s, 3H); ^13^C-NMR (500 MHz, DMSO-*d*_6_) δ 171.76, 167.31, 164.23, 148.63, 143.34, 139.74, 137.98, 135.55, 133.66, 132.97, 131.28, 131.08, 129.43, 129.28, 127.40, 126.19, 126.12, 126.04, 124.74, 18.01; HRMS calcd. for C_21_H_13_Cl_3_N_4_O_2_ [M + Na]^+^ 480.9996, found 480.9995.

*N-(4-(tert-butyl)phenyl)-2-chloro-5-(5-(3,6-dichloropyridin-2-yl)-1,2,4-oxadiazol-3-yl)benzamide***7e**. Grey solid, yield 69%, m.p. 208–210 °C, ^1^H-NMR (500 MHz, DMSO-*d*_6_)δ 10.64 (s, 1H), 8.35 (d, *J* = 9.0 Hz,1H), 8.23–8.18 (m, 2H), 7.92 (d, *J* = 8.5 Hz, 1H), 7.83 (d, *J* = 9.0 Hz, 1H), 7.64 (d, *J* = 8.5 Hz, 2H), 7.38 (d, *J* = 8.5 Hz, 2H), 1.28 (s, 3H); ^13^C-NMR (500 MHz, DMSO-*d*_6_) δ 171.70, 167.25, 163.71, 148.58, 146.49, 143.31, 139.67, 137.91, 136.12, 133.65, 131.24, 131.05, 129.41, 129.24, 127.28, 126.44, 125.44, 124.71, 119.54, 34.09, 31.18; HRMS calcd. for C_24_H_19_Cl_3_N_4_O_2_ [M + Na]^+^ 523.0466, found 523.0466.

*2-chloro-5-(5-(3,6-dichloropyridin-2-yl)-1,2,4-oxadiazol-3-yl)-N-(3-(trifluoromethyl)phenyl)benzamide***7f**. White solid, yield 77%, m.p. 191–195 °C, H-NMR (500 MHz, DMSO-*d*_6_) δ 11.02 (s, 1H), 8.30 (d, *J* = 8.5 Hz, 1H), 8.25 (s, 1H), 8.19 (s, 2H), 7.87 (d, *J* = 8.5 Hz, 2H), 7.81 (d, *J* = 8.5 Hz, 1H), 7.58 (t, 1H), 7.45 (d, *J* = 7.5 Hz, 1H); ^13^C-NMR (500 MHz, DMSO-*d*_6_) δ 171.76, 167.24, 164.35, 148.61, 143.32, 139.67, 139.42, 137.30, 133.66, 131.27, 131.15, 130.19, 129.81, 129.26, 127.45, 124.84, 123.37, 120.48, 115.84; HRMS calcd. for C_21_H_10_Cl_3_F_3_N_4_O_2_ [M + Na]^+^ 534.9714, found 534.9717.

*2-chloro-5-(5-(3,6-dichloropyridin-2-yl)-1,2,4-oxadiazol-3-yl)-N-(2,4-dimethylphenyl)benzamide***7g**. Yellow solid, yield 60%, m.p. 225–226 °C, ^1^H-NMR (500 MHz, DMSO-*d*_6_) δ 10.09 (s, 1H), 8.37 (d, *J* = 8.5 Hz, 1H), 8.25 (d, *J* = 2.0 Hz, 1H), 8.22 (d, *J* = 2.0 Hz,1H), 8.20 (d, *J* = 2.0 Hz, 1H), 7.94 (d, *J* = 8.5 Hz, 1H), 7.84 (d, *J* = 8.0 Hz, 1H), 7.35 (d, *J* = 8.0 Hz, 1H), 7.09 (s, 1H), 7.04 (d, *J* = 8.0 Hz, 1H), 2.29 (s, 3H), 2.27 (s, 3H); ^13^C-NMR (500 MHz, DMSO-*d*_6_) δ 171.76, 167.31, 164.23, 148.62, 143.34, 139.75, 138.05, 135.33, 133.65, 132.94, 132.83, 131.27, 131.07, 130.98, 129.37, 129.28, 127.38, 126.61, 125.99, 124.71, 20.56, 17.92; HRMS calcd. for C_22_H_15_Cl_3_N_4_O_2_ [M + Na]^+^ 495.0153, found 495.0154.

*2-chloro-5-(5-(3,6-dichloropyridin-2-yl)-1,2,4-oxadiazol-3-yl)-N-(2-fluorophenyl)benzamide***7h**. Yellow solid, yield 60%, m.p. 203–207 °C, ^1^H-NMR (500 MHz, DMSO-*d*_6_) δ 10.57 (s, 1H), 8.35 (d, *J* = 8.5 Hz, 1H), 8.27–8.19 (m, 2H), 7.92 (d, *J* = 8.5 Hz, 1H), 7.90–7.86 (m, 1H), 7.83 (d, *J* = 8.0 Hz, 1H), 7.35–7.21 (m, 3H); ^13^C-NMR (500 MHz, DMSO-*d*_6_) δ 171.78, 167.30, 164.41, 148.65, 143.34, 139.73, 137.35, 133.79, 131.29, 131.10, 129.65, 129.29, 127.54, 126.81 (d, *J* = 7.5 Hz), 125.70, 125.28, 125.19, 124.69, 124.51 (d, *J* = 3.0 Hz), 115.88 (d, *J* = 19.0 Hz); HRMS calcd. for C_20_H_10_Cl_3_FN_4_O_2_ [M + Na]^+^ 484.9746, found 484.9747.

*2-chloro-5-(5-(3,6-dichloropyridin-2-yl)-1,2,4-oxadiazol-3-yl)-N-(3-fluorophenyl)benzamide***7i**. Yellow solid, yield 62%, m.p. 228–230 °C, ^1^H-NMR (500 MHz, DMSO-*d*_6_) δ 10.91 (s, 1H), 8.35 (d, *J* = 9.0 Hz, 1H), 8.30–8.20 (m, 2H), 7.92 (d, *J* = 8.5 Hz, 1H), 7.86 (d, *J* = 8.5 Hz, 1H), 7.70 (d, *J* = 11.5 Hz, 1H), 7.53–7.36 (m, 2H), 7.03–6.92 (m, 1H); ^13^C-NMR (500 MHz, DMSO-*d*_6_) δ 171.76, 167.23, 164.15, 163.10, 161.18, 148.61, 143.33, 139.69, 137.46, 133.62, 131.21 (d, *J* = 16.0 Hz), 130.58 (d, *J* = 9.5 Hz), 129.72, 129.27, 127.37, 124.82, 115.55, 110.64 (d, *J* = 21.0 Hz), 106.68, 106.47; HRMS calcd. for C_20_H_10_Cl_3_FN_4_O_2_ [M + Na]^+^ 484.9746, found 484.9747.

*2-chloro-5-(5-(3,6-dichloropyridin-2-yl)-1,2,4-oxadiazol-3-yl)-N-(4-fluorophenyl)benzamide***7j**. Yellow solid, yield 63%, m.p. 248–251 °C, ^1^H-NMR (500 MHz, DMSO-*d*_6_) δ 10.77 (s, 1H), 8.36 (d, *J* = 8.5 Hz, 1H), 8.26–8.19 (m, 2H), 7.93 (d, *J* = 9.0 Hz,1H), 7.85 (d, *J* = 8.5 Hz, 1H), 7.79–7.72 (m, 2H), 7.22 (t, *J* = 9.0 Hz, 2H); ^13^C-NMR (500 MHz, DMSO-*d*_6_) δ 171.79, 167.29, 163.85, 148.64, 143.36, 139.73, 137.70, 133.68, 131.30, 131.16, 129.62, 129.30, 127.37, 124.82, 121.67 (d, *J* = 7.9 Hz), 115.61, 115.44; HRMS calcd. for C_20_H_10_Cl_3_FN_4_O_2_ [M + Na]^+^ 484.9746, found 484.9749.

*2-chloro-N-(2-chlorophenyl)-5-(5-(3,6-dichloropyridin-2-yl)-1,2,4-oxadiazol-3-yl)benzamide***7k**. Yellow solid, yield 65%, m.p. 195–198 °C, ^1^H-NMR (500 MHz, DMSO-*d*_6_) δ 10.47 (s, 1H), 8.36 (d, *J* =9.0 Hz, 1H), 8.30 (s, 1H), 8.23 (d, *J* = 8.5 Hz, 1H), 7.93 (d, *J* =9.0 Hz, 1H), 7.84 (d, *J* = 8.5 Hz, 1H), 7.75 (d, *J* = 8.0 Hz, 1H), 7.57 (d, *J* = 8.0 Hz, 1H), 7.42 (t, *J* = 7.5 Hz, 1H), 7.32 (t, *J* = 7.5 Hz, 1H); ^13^C-NMR (500 MHz, DMSO-*d*_6_) δ 171.76, 167.27, 164.43, 148.64, 143.34, 139.73, 137.27, 134.23, 133.81, 131.27, 131.15, 129.75, 129.66, 129.28, 128.60, 127.87, 127.70, 127.61, 127.58, 124.68; HRMS calcd. for C_20_H_10_Cl_4_N_4_O_2_ [M + Na]^+^ 500.9450, found 500.9452.

*2-chloro-N-(3-chlorophenyl)-5-(5-(3,6-dichloropyridin-2-yl)-1,2,4-oxadiazol-3-yl)benzamide***7l**. Yellow solid, yield 65%, m.p. 214–218 °C, ^1^H-NMR (500 MHz, DMSO-*d*_6_) δ 10.90 (s, 1H), 8.35 (d, *J* = 9.0Hz, 1H), 8.26 (d, *J* = 2.0 Hz, 1H), 8.23 (dd, *J* = 8.5, 2.0 Hz, 1H), 7.96–7.90 (m, 2H), 7.85 (d, *J* = 8.5 Hz, 1H), 7.60 (d, *J* = 9.0 Hz, 1H), 7.41 (t, *J* = 8.5 Hz, 1H), 7.22 (s, 1H); ^13^C-NMR (126 MHz, DMSO-*d*_6_) δ 171.79, 167.27, 164.22, 148.64, 143.36, 140.09, 139.71, 133.67, 133.23, 131.31, 131.18, 130.66, 129.78, 129.30, 127.42, 124.86, 123.91, 119.30, 118.24; HRMS calcd. for C_20_H_10_Cl_4_N_4_O_2_ [M + Na]^+^ 500.9450, found 500.9454.

*2-chloro-N-(4-chlorophenyl)-5-(5-(3,6-dichloropyridin-2-yl)-1,2,4-oxadiazol-3-yl)benzamide***7m**. Yellow solid, yield 67%, m.p. 242–243 °C. ^1^H-NMR (500 MHz, DMSO-*d*_6_) δ 10.85 (s, 1H), 8.35 (d, *J* = 8.5 Hz, 1H), 8.28–8.18 (m, 2H), 7.92 (d, *J* = 9.0 Hz, 1H), 7.85 (d, *J* = 8.5 Hz, 1H), 7.76 (d, *J* = 9.0 Hz, 2H), 7.44 (d, *J* = 9.0 Hz, 2H); ^13^C-NMR (126 MHz, DMSO-*d*_6_) δ 171.79, 167.28, 164.03, 148.64, 143.36, 139.72, 137.64, 137.57, 133.67, 131.31, 131.17, 129.71, 129.30, 128.84, 127.80, 127.39, 124.84, 121.39; HRMS calcd. for C_20_H_10_Cl_4_N_4_O_2_ [M + Na]^+^ 500.9450, found 500.9452.

*N-(4-bromophenyl)-2-chloro-5-(5-(3,6-dichloropyridin-2-yl)-1,2,4-oxadiazol-3-yl)benzamide***7n**. Yellow solid, yield 69%, m.p. 242–245 °C, ^1^H-NMR (500 MHz, DMSO-*d*_6_) δ 10.86 (s, 1H), 8.35 (d, *J* = 9.0 Hz, 1H), 8.28–8.19 (m, 2H), 7.92 (d, *J* = 9.0 Hz, 1H), 7.85 (d, *J* = 8.5 Hz, 1H), 7.71 (d, *J* = 9.0 Hz, 2H), 7.57 (d, *J* = 8.5 Hz, 2H); ^13^C-NMR (126 MHz, DMSO-*d*_6_) δ 171.72, 167.22, 163.96, 148.59, 143.29, 139.65, 138.03, 137.52, 133.62, 131.68, 131.24, 131.10, 129.64, 129.24, 127.35, 124.78, 121.67; HRMS calcd. for C_20_H_10_BrCl_3_N_4_O_2_ [M + Na]^+^ 544.8945, found 544.8942.

### 3.3. Biological Activity Test

The insecticidal activity was tested according to [28]. The results of the activity test are shown in Table 2 and Table 3. The fungicidal activity of all the synthetic compounds were tested in vitro against eight fungi using a mycelia growth inhibition method according to references [29,30]. *Alternaria solani* (AS), *FusaHum graminearum* (FG), *Cercospora arachidicola* (CA), *Phytophthora capsica* (PC), *Sclerotinia sclerotiorum* (SS), *Botrytis cinereal* (BC), *Thanatephorus cucumeris* (TC), *Fusarium oxysporum* (FO) were provided by the National Pesticide Engineering Research Centre, Nankai University. The results of the activity test are shown in Table 4.

## 4. Conclusions

Novel benzamides substituted with pyridine-linked 1,2,4-oxadiazole were designed by bioisosterism, and synthesized easily via esterification, cyanation, cyclization and aminolysis reactions. Through using CuCN as the cyanidation reagent, l-proline as the catalyst, and increasing the temperature gradually in the cyanation reaction, we got the best yield (79%) and reduced the risk of this experiment. The structures of the target compounds were confirmed by ^1^H-NMR, ^13^C-NMR and HRMS. The preliminary bioassay results showed that most compounds had good larvicidal activity against mosquito larvae at 10 mg/L, especially compound **7a** with excellent larvicidal activity (100%); even at 1 mg/L, the larvicidal actiity was still 40%; at 50 mg/L, all the target compounds showed good fungicidal activity against the eight tested fungi. Compound **7h** exhibited good inhibitory activity (90.5%) against *Botrytis cinereal*, which was better than fluxapyroxad (63.6%). In addition, it had moderate inhibitory activities against *Alternaria solani* (50.0%), *Sclerotinia sclerotiorum* (80.8%) and *Thanatephorus cucumeris* (84.8%). Therefore, these compounds could potentially be the lead compounds for further optimisation.

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
