# Peer review of "Synthesis and Biological Activity of Benzamides Substituted with Pyridine-Linked 1,2,4-Oxadiazole"

_molecules, 2020, doi:10.3390/molecules25153500_

Round 1
Reviewer 1 Report
The authors present the synthesis of a benzamides series substituted with both pyridine and 1,2,4-oxadiazole rings in six reaction steps starting from 2-chloro-5-iodobenzoic acid, which was designed by bioisosterism with respect some bioactive analogous amides. The synthesis shows an acceptable methodological contribution and in general the method is very simple and practical. In general, the thematic is interesting since the obtained products exhibited good larvicidal activities against mosquito larvae. However, the work is not very well presented and has too many spelling and punctuation mistakes, as well as writing problems, and also, the work is not very well complemented. From these details, I think that this manuscript is appropriate for publication as a full paper in Molecules after major revisions and answers for the questions, suggestions and/or corrections.
- Abstract: Some important synthetic detail should be included.
- Keywords must be in alphabetical order.
- Introduction: Check various spelling and commas mistakes. Scheme 1: More important information should be included, e.g., Yield and time of the reactions. Likewise, describe the substituent group R like this: 7a-n, R: 7a, H; 7b, 2-CH3; 7c, 3-CH3; etc.
- Results and Discussion: Check various spelling mistakes. In general, authors should carry out a better and complete analysis, for example, in the transformation from 4 to 5, the intermolecular cyclocondensation reaction between 4 and pyridine acid is favored instead of the reaction of 4 with itself? etc.
- Experimental section: General information-Change ‘1HNMR and 13CNMR’ to ‘1H NMR and 13C NMR’; Synthesis-Capitalize the first letter of names in subtitles and when the paragraph begins with the compound name. Review the score of all reported 13C NMR data because there are several spaces left over. HRMS analysis is wrong for 7f and 7g, 7f is ‘523.9714/523.9717’ and must be ‘534.9714/534.9717’, while 7b is ‘495.0513/495.0514’ and must be ‘495.0153/495.0154’; Biological activity test-Italicize 'in vitro' and check commas; Conclusions-Correct spelling and include a synthetic conclusion.
Reviewer 2 Report
Manuscript ID: molecules-883984
Title: Synthesis and Biological Activity of Benzamides Substituted with Pyridine-linked 1,2,4-oxadiazole
This paper reports the preparation and the insecticidal and fungicidal activity of new heterocyclic compounds in order to assess their potential use as new synthetic pesticides. From a synthetic point of view, there is no original contribution except a further generalization of the synthetic steps used, in particular the second one, but it could be quite interesting for reported bioassays and as a further study on molecular structure-activity correlations. I think that it could be publishable in Molecules, but only after a careful revision of the entire manuscript. Anyway, I have to do some observations which should be taken in to account for the preparation of a revised manuscript
First, the whole text should be carefully reviewed with regard to language and form.
Introduction: the cited literature appears “one color”. At least the most recent reviews on biological activities of heterocyclic compound should be mentioned.
The second step of the synthesis, i.e. the CuCN-mediated cyanation, is an L-proline promoted Rosenmund-von Braun reaction, according to K. Ding et al. in Synlett 2008, 69-72 (see also Eur. J. Med. Chem. 2016, 118, 170-177), in which an application of this reaction more similar to that described in the manuscript is reported, since it concerns the transformation of an aryl iodide into aryl cyanide. These two articles should be cited rather than references 22 and 23 of the text.
Also, it is right that authors describe the optimization of reaction conditions for the step 2, but yields, in Table 1, must be expressed as whole numbers.
Line 62: compound 3 is 5-cyano not 5-iodo derivative.
Line 114: Table 2 is unclear; results on mosquito larvae should be reported in a different table
Finally, the results obtained in biological tests deserve a deeper discussion.
Round 2
Reviewer 1 Report
The revised manuscript delivered all the comments in right way. Now it's good enough to be accepted for publication in Molecules.
Some details:
- Some capital letters are missing.
- Supporting information has titles in another language.
Reviewer 2 Report
The revised form of the manuscript seems acceptable to me. Only, chemical yields should be indicated as whole numbers anywhere in the text